# A Novel Microelectrode Based on Joule Heating and Impedance Spectroscopy for Inducing and Monitoring the Aggregation of HCV-Specific Probes

**DOI:** 10.3390/s25113312

**Published:** 2025-05-24

**Authors:** Reda Abdelbaset, Omar E. Morsy, Mariam Hossam Eldin, Sherif M. Shawky, Yehya H. Ghallab, Yehea Ismail

**Affiliations:** 1Biomedical Engineering Department, Faculty of Engineering, Helwan University, Cairo 11795, Egypt; 2Center of Nanoelectronics and Devices, The American University in Cairo (AUC), Zewail City of Science and Technology, Cairo 11835, Egypt; 3Center for Materials Science, Zewail City of Science and Technology, Giza 12578, Egyptsshawky@zewailcity.edu.eg (S.M.S.); 4Department of Biochemistry, College of Pharmaceutical Sciences and Drug Manufacturing, Misr University for Science and Technology, Giza 12566, Egypt; 5Electronics and Communication Engineering Department (ECNG), The American University in Cairo (AUC), Cairo 11835, Egypt

**Keywords:** hepatitis C virus (HCV), gold nanoparticles, aggregation, Joule heating, impedance spectroscopy

## Abstract

**Highlights:**

**What are the main findings?**
A new on-chip microelectrode was designed to induce and monitor the aggregation of HCV RNA-attached gold nanoparticles (AuNPs).The microelectrode effectively induces the aggregation of RNA-attached AuNPs through controlled Joule heating.Impedance spectroscopy was utilized to monitor the aggregation process in real time.Integrating Joule heating and impedance spectroscopy enables on-chip system integration and reduces the detection time for nanoparticle aggregation, enhancing its bioanalytical utility.

**What are the implications of the main findings?**
The developed microelectrode can heat a droplet to a specific temperature.It successfully induces color change based on the presence of hepatitis C virus (HCV) RNA.Impedance readings are used to monitor the aggregation process.

**Abstract:**

The world urgently needs new methods to quickly and efficiently detect mutated viruses. An RNA-AuNP-based colorimetric biosensor is a highly sensitive, specific, and cost-effective tool that enables rapid, visual detection of target molecules for applications in disease diagnostics, environmental monitoring, and forensic analysis. An RNA-AuNP-based colorimetric biosensor requires precise control over nanoparticle dispersion and aggregation, which can be achieved using temperature regulation. A novel on-chip microelectrode is proposed to induce and monitor the aggregation of RNA-attached gold nanoparticles (AuNPs) through Joule heating and impedance spectroscopy. The proposed platform is implemented based on printed circuit board (PCB) technology, which has many advantages, such as fast and easy design and fabrication, low power consumption, and low costs. Joule heating is the process in which the energy of an electric current is converted into heat as it flows through a resistance. Impedance spectroscopy is an analytical technique that measures a system’s electrical response to an applied AC signal across a range of frequencies, providing insights into a sample’s dielectric properties. The results validate that the fabricated microelectrode is capable of heating a 20 µL droplet to 75 °C within 30 s, utilizing a low power input of only 3.75 watts and successfully inducing a color change based on the presence of hepatitis C virus (HCV) RNA, while impedance readings are used to monitor the aggregation.

## 1. Introduction

Bacteria and viruses have evolved into formidable adversaries, continuously adapting to evade traditional diagnostic and therapeutic approaches. Their ability to mutate, develop resistance, and exploit host mechanisms makes them increasingly difficult to detect and treat effectively [1]. Therefore, developing new methods for detecting these pathogens in an effective and fast way is crucial. In 2024, approximately 50 million people worldwide were living with HCV, with an estimated 290,000 deaths annually due to HCV-related complications [2].

Recent advances in viral detection have taken advantage of the unique optical features of gold nanoparticles (AuNPs) to provide fast and sensitive diagnostic procedures [3,4]. Gold nanoparticles (AuNPs) are tiny gold particles with a 1 to 100 nm diameter, also known as colloidal gold [5]. Gold nanoparticles (AuNPs) are extensively used in biosciences for applications such as diagnostics, tissue regeneration, drug delivery, biosensing, bioimaging, and vaccine development due to their unique optical and electronic properties [4,6,7,8]. One such technique includes the aggregation of AuNPs upon engagement with viral targets, resulting in a noticeable color shift that allows for straightforward recognition [9].

Gold nanoparticles (AuNPs) can be aggregated using various techniques depending on the desired application, such as salt-induced [10], pH-induced [11], biomolecule-induced [12], solvent-induced [13], enzyme-mediated [14], and temperature-induced aggregations [15]. Temperature-induced aggregation of AuNPs offers reversibility, high specificity, a rapid response, and precise external control without requiring chemical additives, making it a cleaner and more controllable alternative to other aggregation methods [16,17].

Temperature-induced aggregation of gold nanoparticles (AuNPs) is controlled by precisely regulating the temperature and nanoparticle properties, using methods such as direct heating and cooling, thermo-responsive polymer coatings, biomolecular denaturation, plasmonic photothermal heating, and ionic/solvent environment regulation [5,18]. Direct heating and cooling, especially Joule heating, offer precise, rapid, and reversible temperature control without requiring chemical modifications. This ensures high reproducibility, uniform aggregation, cost-effectiveness, and scalability compared to other temperature-induced aggregation techniques, such as water baths and infrared lasers [19].

Joule heating offers significant advantages over water bath and laser methods for RNA aggregation, particularly in terms of efficiency, precision, and scalability. Unlike water baths, which suffer from a slow thermal response and evaporation issues, Joule heating provides rapid and uniform temperature control, minimizing sample degradation. Compared to laser heating, which can create localized hotspots and requires complex optical calibration, Joule heating ensures consistent thermal distribution without the risk of RNA fragmentation. Additionally, its direct electrical resistance mechanism allows for fine-tuned temperature adjustments, making it highly adaptable for various RNA studies [19,20]. Joule heating, also known as resistive heating, is the conversion of electrical energy into heat when current flows through a resistive material following Joule’s law, which states that heat generation is proportionate to the square of the current, resistance, and time [21]. In the existing literature, the limited number of works utilizing this model faced several challenges, including the need for a high voltage and the use of alternative heating technologies like lasers. Malekanfard, Amirreza, et al. developed a system for Joule heating-induced electrothermal flow circulations to entrain nanoparticles and enrich 0.5 µm diameter particles [22]. Zhang, Lin, et al. designed a method to synthesize hollow Au spheres by rapidly heating and cooling carbon-supported gold nanoparticles [21]. Liu, Xiaoming, et al. demonstrated an electric treatment of mono-dispersed particles to create aggregated AuNPs and investigated their bulk heating behavior under a 655 nm laser and a 13.56 MHz RF electric field [23].

Impedance spectroscopy is an analytical technique that measures a material’s electrical response to an alternating current (AC) signal across frequencies, providing insights into its resistance, capacitance, and conductivity [24]. Impedance spectroscopy is an efficient, label-free, rapid, and extremely sensitive technology that allows for real-time, multi-frequency investigations of resistive and capacitive characteristics [24]. MacKay, Scott, et al. implemented a platform to characterize surfaces that were modified with gold nanoparticles using impedance spectroscopy [25]. Zhou, Yaping, et al. created a label-free electrochemical impedance spectroscopy sensor using gold-coated magnetic nanoparticles to detect vascular endothelial growth factor in cell culture media [26].

In this work, a novel dual-function microelectrode (DFM) for initiating and monitoring AuNP aggregation using Joule heating and impedance spectroscopy was designed, evaluated, simulated, and implemented to detect HCV RNA. The Joule heating-based microelectrode is proposed to induce AuNP aggregation through direct heating at a certain temperature. Furthermore, the aggregation process of the AuNPs was monitored by impedance spectroscopy using the same microelectrode by periodically switching it from heating mode to impedance measuring. Sequencing the two modes is preferred to prevent interference between them and enable the use of the same dual-function microelectrode. This approach enhances the compactness and efficiency in both modes by minimizing excessive spacing, which can affect impedance measurements and the uniformity of the heat generation. The DFM was designed based on PCB technology according to its capabilities, offering benefits such as easy repair, low electronic noise, cost-effectiveness, high reliability, and potential use as a diagnostic tool [27]. The proposed platform was analyzed using a mathematical derivative to calculate resistance and study heat generation factors, while the heat produced by the DFM was simulated with the finite element method (FEM) in COMSOL Multiphysics, version 6. The whole system was tested using probe and synthetic RNA targets to prove the ability of the DFM to induce and state the aggregation of AuNPs.

## 2. Materials and Methods

A novel technique leveraging the Joule heating phenomenon and impedance spectroscopy for inducing and monitoring the aggregation of RNA-attached AuNPs was implemented using PCB technology, as illustrated in Figure 1. During the AuNP aggregation process, the sample color transitions from red to violet due to particle binding [28].

This phenomenon allows for the detection of various small molecules, including cysteine, homocysteine, trinitrotoluene, melamine, cocaine, Adenosine triphosphate (ATP), glucose, and dopamine [28]. SM Shawky et al. developed a nanoprobe by functionalizing citrate-capped AuNPs with a thiolated HCV-specific probe, which was then mixed with the RNA sample and heated at 95 °C for 3 min [29]. Figure 1 also illustrates the proposed platform, which comprises two main stages: the heating mode, consisting of a power supply and DFM, and the impedance spectroscopy mode, which includes the same DFM along with an impedance spectroscopy system, both of which will be explained in the next section. In addition, the entire system is managed by a microcontroller, which controls the switching between two modules and regulates the applied voltage for heating.

### 2.1. The Joule Heating Microelectrode (JHM)

Joule heating is the process by which the passage of an electric current through a conductor produces heat [30]. The equation that describes the generation of heat due to Joule heating is as follows [31]:(1)$$Q=I^{2}\times R$$
where *Q* is the quantity of heat, and *I* is the applied current to the microelectrode. *R* is the resistance of the electrode. *t* is the time of applying the electrical current to the microelectrode. The previous equation can be modified to convert the quantity of heat from watts to joules [31].(2)$$Q=I^{2}\times R\times t$$

The specific heat capacity is a thermal index, expressed by *C*, as seen in the following Equation [32]:(3)Q=C×m×∆T(4)$$m=m_{0}\times V$$(5)$$∆T=\frac{I^{2}\times R\times t}{C\times m_{0}\times V}$$

*C* is the heat capacity of the material of the microelectrode. *M* is the mass of the microelectrode. *M*_0_ is the mass per cubic centimeter of the fabric of the microelectrode. *V* is the volume of the microelectrode. Therefore, it is possible to deduce the factors affecting the warming of the electrodes as shown in the final equation of the temperature change. However, these factors affected as follows:The higher the resistance is, the higher the temperature of the microelectrode is.The higher the applied current is, the higher the temperature of the microelectrode is.The longer the current is applied for, the higher the temperature of the microelectrode is.The lower the volume of the microelectrode is, the higher the temperature of the microelectrode is.

### 2.2. Impedance Spectroscopy

Impedance spectroscopy measures the resistance and capacitance properties by applying a sinusoidal AC excitation signal [24]. The impedance spectrum is obtained by changing the frequency over a wide range of AC signals. The following are the governing equations for calculating dielectric properties depending on the amplitude of input/output signals and the phase shift between them [24]:(6)$$x_{1}=\frac{-A_{i}R_{f}}{A_{o}\sqrt{1+{\left(2\pi fC_{f}R_{f}\right)}^{2}}}$$(7)$$x_{2}=\frac{-A_{i}R_{f}}{A_{o}\sqrt{1+{\left(2\pi fC_{f}R_{f}\right)}^{2}}}$$(8)$$Q_{1}=-ph-\tan^{-1}\left(2\pi fC_{f}R_{f}\right)$$(9)$$Q_{2}=-\tan^{-1}\left(2\pi fC_{f}R_{f}\right)$$(10)$$R_{z}=x_{1}\cos\left(Q_{1}\right)+x_{2}\cos\left(Q_{2}\right)$$(11)$$I_{z}=x_{1}\sin\left(Q_{1}\right)+x_{2}\sin\left(Q_{2}\right)$$(12)$$impedance\;maggnitude\;\left|Z\right|\left(Ω\right)=\sqrt{R_{z}^{2}+I_{z}^{2}}$$
where *A_i_* is the amplitude of the applied signal, *R_f_* is the resistance of the feedback resistor of the readout circuit, *A_o_* is the amplitude of the output signal of the readout circuit, *f* is the frequency of the applied signal, *C_f_* is the capacitance of the feedback capacitor of the readout circuit, and *ph* is the phase shift between the applied and output signal.

### 2.3. Finite Element Method (FEM)

An FEM model was implemented using COMSOL Multiphysics to simulate the proposed PCB-based DFM in different actual conditions (ambient temperature and sample effect). Furthermore, the FEM model confirmed the credibility of the suggestions of the mathematical model. The proposed model was developed based on the following hypothesis: a three-dimensional (3D) space domain is employed to enhance the accuracy of the simulation. Two physical phenomena are considered to simulate the Joule heating process: the *electric current* module, which models the effect of the applied current, and the *heat transfer in solids* module, which simulates the transfer of temperature through the system by convection. Both physics equations are solved in a time-dependent manner to analyze their temporal effects. Temperature coupling and an electromagnetic heat source are incorporated to establish the interaction between the two physical domains. However, four significant steps are required to perform the FEM model, which are detailed in the following sections.

#### 2.3.1. Geometry

The structure of each design was built using COMSOL tools according to the specifications of the proposed fabrication technology, as shown in Figure 2. The width and the length of the proposed design of the DFM were selected based on the possibilities of PCB technology. The spiral shape was suggested for many reasons. The most important one is that it allows for the design of a long electrode in a small area. Figure 2 shows the geometry of 5 turns of the DFM with 100 µm width and space intervals based on PCB technology. FR-4 (standard material) was proposed as a substrate material. However, the dimensions were optimized to align with the manufacturing capabilities of printed circuit board (PCB) technology. In addition, droplets with three distinct contact angles were simulated to investigate the effects of the contact angle and surface area of the droplet on the heating system’s behavior.

As shown in Table 1, the length and resistance of the DFM were estimated to analyze them. They can be calculated as follows: The length of the microelectrode is measured using Altium Design 20 for PCB technology. The resistance of the microelectrode is estimated by the following: The mathematical method using the following equation, where *R* is the resistance of the microelectrode, ρ is the material resistivity (ρ of copper equal $1.72\times 10^{-8}$ [33]), *l* is the length of the microelectrode, and *A* is the area of the microelectrode (equal to the width in case of ignoring the thickness of the microelectrode):(13)$$R=\rho \frac{l}{A}\;\rho =1.72\times 10^{-8}$$The finite element method: The resistance is estimated by applying a voltage difference to a conductor to create a current flow. The intensity of the current is usually a function of the applied voltage difference. In the most straightforward (linear) case, the current flow and the voltage difference are proportional; the proportionality constant is the device’s resistance. The resistance of each design geometry is estimated by solving the electric current module (ec) based on a stationary study.

#### 2.3.2. Material Properties

To simulate the Joule heating and impedance spectroscopy, certain material properties are needed to solve the governing equations of the two applied modules (electric current and heat transfer in solids). For the electric current module, these are electrical conductivity and relative permittivity. For heat transfer in solids, these are density, thermal conductivity, and heat capacity. The copper and FR-4 are proposed for the DFM and substrate, respectively. The saline was selected as a droplet (as an example).

#### 2.3.3. Loads and Boundary Conditions

Many tests were performed to study the DFM under different conditions as follows: Dry test: The DFM was tested at an ambient temperature, which was represented by the convection heat flux. The convection heat flux of air was described by the heat transfer coefficient as 5 [W/(m^2^·K)], and the ambient temperature is set to 20 °C.Sample test: A droplet of saline was used as an example in a half-sphere shape. A 20 µL drop is preferred for PCB technology to be suitable for the heat generated by the DFM.Cooling test: An initial 1 s duration pulse was applied to heat the DFM to 95 °C. Then, it was turned off to study the cooling rate of the DFM at room temperature for an additional four seconds.

#### 2.3.4. Meshing

The proposed model was meshed using a normal element size, which was selected based on a grid independence study, as detailed in Table 2. As shown in Table 2, the normal grid size required less computational time and achieves convergence within 1%, which is a widely accepted criterion for grid independence.

### 2.4. Experimental Setup

A platform, including the DFM chip, readout circuit, and software for controlling and analyzing the data, was established utilizing PCB technology.

#### 2.4.1. DFM Chip

The proposed design of the DFM chip was implemented and fabricated based on PCB technology, utilizing version 20.0.2 of Altium Designer software, as illustrated in Figure 3. The designed chip features a DFM that is printed on the top layer, represented by red-colored wires. Additionally, the DFM is connected to four bonding pads through connection wires, enabling external connections. These connection wires are printed on the bottom layer and are represented by blue-colored wires. An external temperature sensor is used as feedback to adjust the temperature, as shown in Figure 3.

#### 2.4.2. Control System

The control system is designed to perform multiple functions to induce and monitor AuNP aggregation. It includes a switching circuit that enables a seamless transition of the DFM between heating mode and monitoring mode, utilizing two electronically controlled relays. During the heating mode of the control system, a power supply (Rigol DP832) feeds the DFM with the power needed to generate the appropriate heat. During the monitoring mode of the control system, the proposed impedance spectroscopy system consists of major components, as shown in Figure 3, as follows:Function generator: The function generator is implemented using Analog Devices AD9851 synthesizers. The AD9851 digital synthesizer generates the required signals with a 5 MHz bandwidth.The design features wrapped electrodes with dual synchronized functions (Figure 3). A DC signal heats the sample, while a control circuit with two relays switches between heating and impedance microscopy modes during the pulsed signal’s off time, enabling real-time aggregation monitoring.Readout circuit: The readout circuit converts the impedance of the microelectrode to a voltage signal using a charge amplifier, as shown in Figure 4 [24].Data acquisition: The signal from the readout circuit is acquired using Saleae Logic 8 with appropriate features up to a 5 MHz frequency. Table 3 shows the specifications for the Saleae Logic Pro 8 data acquisition [34].
Temperature sensor: The generated temperature was measured using a Fluke 87 V Industrial Multimeter, which was pre-calibrated by the manufacturer. According to the manufacturer’s specifications, the multimeter is capable of measuring the temperature with an error margin of ±2 °C at an input of 100 °C and ±1 °C at 0 °C [35].Software plays a crucial role in estimating impedance, serving as the final step in the process. Various tasks must be executed to achieve accurate impedance estimation. In version 17.0 of LabVIEW, a user interface is designed to read the input AC signal from the function generator and the output AC signal from the readout circuit. It determines the phase shift between the input and output signals and exports the amplitude and phase shift data in an Excel format. Additionally, Version R2023b of MATLAB was utilized to estimate impedance by implementing a script based on the transfer function of the readout circuit, which is defined as follows:
(14)$$\left|z\right|=\frac{A_{i}\times R_{f}}{A_{o}\times \sqrt{1+2\pi f\times C_{f}\times R_{f}}}$$
(15)$$∠z={\mathrm{Tan}}^{-1}\;\left((2\pi f\times C_{f}\times R_{f}\right)+ph+180)$$
where $\left|z\right|\;and\;∠\;z$ are the magnitude and angle of the microelectrode impedance. $A_{i}\;and\;A_{o}$ Are the amplitudes of the input and output signals. ph is the phase shift between the input and output signals. $C_{f}\;and\;R_{f}$ are the capacitance and resistance of the feedback of the charge amplifier.

#### 2.4.3. Preparing Sample

##### Nanoprobe

Citrate-capped AuNPs were synthesized according to the conventional reduction method using trisodium citrate and gold chloride trihydrate. The particles were characterized using a UV-Vis device to ensure the synthesis of nanoparticles and determine their size. The next step was the functionalization of an HCV-specific alkanethiol-modified RNA probe on the surface of the citrate AuNPs following the salt aging process [36]. This probe is complementary to the highly conserved HCV RNA 5′-untranslated region (5′UTR) and comprises 32 bases [29].

The functionalization procedure was precisely described by Hill and Mirkin [36]; however, modifications were made to optimize the surface coverage while maintaining nanoparticle stability. This nanoprobe was also characterized using UV-Vis, and the two reagents’ spectra were superimposed, as shown in Figure 5. The functionalization of citrate AuNPs is obvious due to the shift in the spectrum of the nanoprobe toward the longer wavelength, which implies a larger size.

##### Synthetic Targets

Synthetic targets, which were complementary to HCV-specific probes of 100 nucleotides in length, were utilized. Each target was diluted in nuclease-free water to a concentration of 10 pmol/µL to facilitate hybridization with the nanoprobe. However, the nucleotide length (100 nucleotides) and concentration (10 pmol/µL) were optimized to achieve a significant difference in impedance spectroscopy measurements and to ensure the capability of inducing aggregation. Moreover, nonspecific probes were used as a negative control for the experiment to detect AuNP aggregation in the absence of the probe–target complex.

## 3. Results

### 3.1. Simulation

Figure 6A,B illustrate the temperature distribution across the DFM resulting from the application of a 2-ampere current for one second, under dry and sample test conditions, respectively. Unfortunately, the sample affected the generated temperature, as it decreased significantly from 94.5 to 41.5 after one second, as shown in Figure 6. Therefore, the power should be readjusted to reach the desired temperature when heating a sample. Also, Figure 6B,C show that the temperature across the drop sample decreases significantly as it approaches the top. Figure 7 shows the generated temperatures versus time at the center point of the DFM under three different conditions (dry, sample, and cooling tests). The performance of the DFM is significantly affected in the case of the sample test. The generated heat in the presence of a sample is much lower than in the absence of a sample. Therefore, having a sensor to measure the sample temperature and reset the power that is applied to the DFM is important. The cooling test shows the cooling rate of the DFM after being heated using a 1 s pulse.

Figure 8 presents a comparison of different contact angles to investigate their effect on the heating profiles. As illustrated in Figure 8, a decrease in the contact angle results in a reduction in droplet height and an increase in the temperature profile over time, which is consistent with the findings presented in Figure 6. Therefore, minimizing the contact angle as much as possible is crucial.

### 3.2. Experimental Results

Figure 9 shows different AuNP samples (AuNPs, probe + nonspecific RNA, and probe + target RNA) on the top of the PCB before and after heating using the DFM. It can be noticed that the color of each sample was affected as follows [18,37,38]:In the case of AuNPs alone, the sample color changed from red to transparent, which indicates that the nanoparticles remained dispersed and did not aggregate, which is a crucial factor for the observed color change.In the case of the probe + nonspecific RNA sample, the sample color changed from red to violet, indicating AuNP aggregation caused by nonspecific target interactions.In the case of the probe + target HCV RNA sample, the sample color remained unchanged. This stability is attributed to the specific interaction between the AuNPs and the target RNA, preventing aggregation.

The simulation results revealed that heat calculations must consider sample-induced cooling, which was confirmed by a significant temperature drop during the analysis. To achieve sufficient practical heating rates to induce RNA aggregation without causing sample dehydration, the maximum feasible voltage was determined through extensive experimental trials that involved various voltage levels and heating durations. Based on these findings, the application of 2.5 V is recommended to reach a temperature of 75 °C within 30 s, as shown in Figure 10.

The aggregation behaviors of various AuNP samples, including the probes with nonspecific probes and 100-specific HCV RNA probes, were analyzed using spectroscopy, as illustrated in Figure 11. The results demonstrate that the impedance of both samples decreases over time during heating. Furthermore, impedance spectroscopy effectively differentiates between samples containing nonspecific and 100-specific HCV RNA at each stage of the heating process. However, each cycle takes two minutes, including 30 s of heating and 90 s of impedance measuring. A reliability analysis was conducted using the test–retest and Cronbach’s alpha methods to assess the consistency of the heating cycles over time for both the nonspecific and specific RNA. The results demonstrate exceptionally high reliability, with Pearson’s correlation coefficients exceeding 0.99 and Cronbach’s alpha values greater than 0.99 for both the nonspecific and specific RNA. These findings confirm the stability of the measurements across heating cycles. The two-way ANOVA results reveal statistically significant differences in measurements between the nonspecific and specific groups (*p* < 0.001), across heating cycles (*p* = 0.0013), as well as in their interaction (*p* < 0.001). These findings indicate that although both groups exhibit significant degradation with each heating cycle, the specific group initially demonstrates higher values but deteriorates at a faster rate than the nonspecific RNA.

Unlike conventional systems, our innovative approach allows for efficient heat generation using minimal power within a significantly reduced timeframe, as described in Table 4. Besides its smart design with minimal dimensions, it can heat using a small sample size.

Compared to traditional methods such as laser-induced, salt-induced, and water-bath-based aggregation, the proposed Joule heating approach demonstrates clear superiority in several key aspects. Most notably, it achieves aggregation in just 2 min, which is significantly faster than the 30–40 min that are required for laser-based methods and the highly variable times for salt- and water-based methods. Additionally, it maintains high portability and high precision, similar to or better than other methods.

As illustrated in Table 5, in terms of power consumption, Joule heating operates at a lower power level (3.75 watts) compared to laser systems (5 watts), making it more energy efficient. Finally, the cost is low, providing a major advantage over the high costs associated with laser aggregation, while outperforming salt and water methods in terms of both consistency and precision.

Thus, the Joule heating method offers an unmatched combination of speed, portability, energy efficiency, precision, and cost-effectiveness, establishing it as a superior alternative to traditional aggregation techniques. Moreover, the integration of a monitoring system via a dual-function microelectrode on the same chip eliminates the need for additional methods, thereby reducing both the cost and system complexity.

## 4. Conclusions

A novel microelectrode-on-chip for inducing and monitoring RNA-AuNP aggregation is presented and discussed. A mathematical and finite element method model was implemented to study the DFM. The proposed DFM was implemented based on PCB technology. Different conditions (dry, droplet, and cooling) were applied to evaluate the proposed DFM. All results (simulation and experimental) confirm the ability of a PCB-based DFM to achieve a certain temperature in a short time. The impedance of DFM was measured under a wide range of frequencies (100 kHz to 3 MHz). The experimental results show the ability of the DFM to induce and characterize RNA-AuNP aggregation efficiently using Joule heating and impedance spectroscopy. The simulation and experimental results demonstrate that the DFM, utilizing PCB technology, is a highly suitable candidate for inducing and monitoring AuNP aggregation and detection of HCV RNA. Furthermore, the experimental results validate the capability of the proposed microelectrode to heat the tested sample to 75 °C within 30 s, using 3.75 watts of power. Leveraging its integrated on-chip design, it offers rapid operation, low power consumption, portability, and ease of use without requiring specialized expertise.

## Figures and Tables

**Figure 1 sensors-25-03312-f001:**
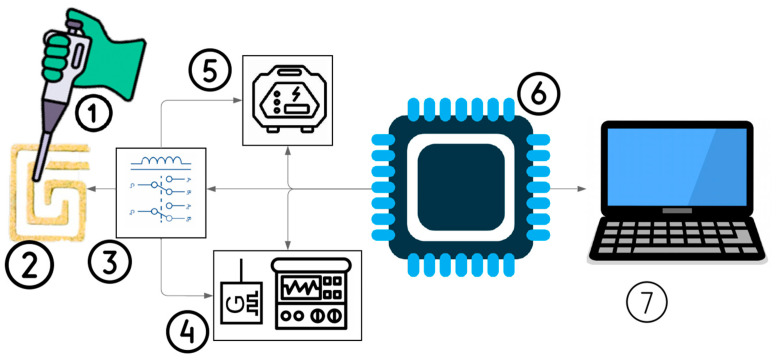
A block diagram of the proposed platform for RNA-AuNP aggregation. (1) RNA-AuNP sample release, (2) DFM, (3) switching circuit between two modes, (4) impedance spectroscopy for monitoring RNA-AuNP aggregation, (5) power supply for heating and inducing RNA-AuNP aggregation, (6) microcontroller for controlling the overall system, and (7) personal computer to analyze the data.

**Figure 2 sensors-25-03312-f002:**
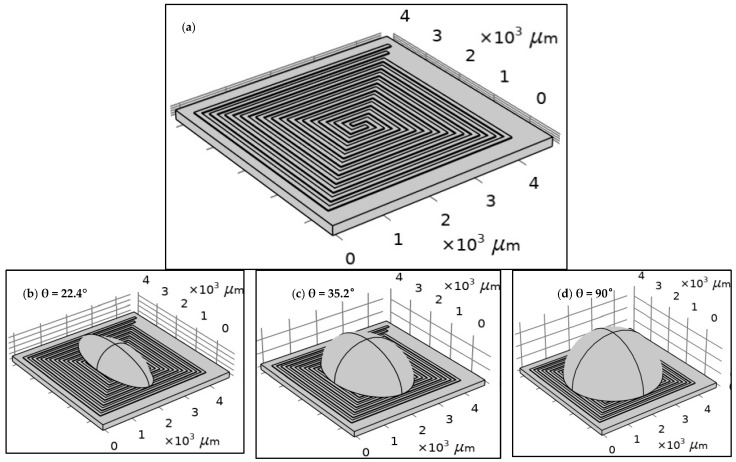
The geometry of the dual-function microelectrode (DFM): (**a**) without a droplet and with a droplet with different contact angles: (**b**) θ = 22.4°, (**c**) θ = 35.2°, and (**d**) θ = 90°. As an initial boundary condition, all surfaces of the chip are exposed to ambient air at a room temperature of 20 °C, with a convective heat transfer coefficient of 5 W/(m^2^·K), and no thermal insulation is applied.

**Figure 3 sensors-25-03312-f003:**
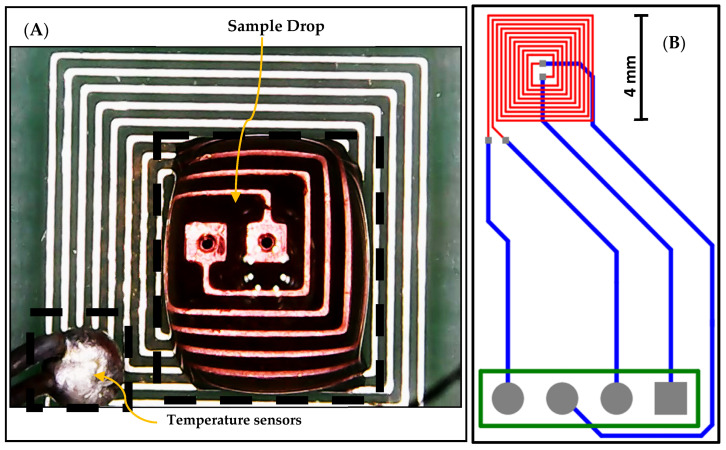
(**A**) The fabricated chip and external temperature sensor, and (**B**) the layout of the fabricated DFM chip based on PCB technology.

**Figure 4 sensors-25-03312-f004:**
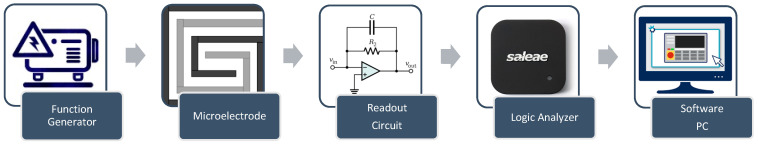
A block diagram of the impedance spectroscopy system.

**Figure 5 sensors-25-03312-f005:**
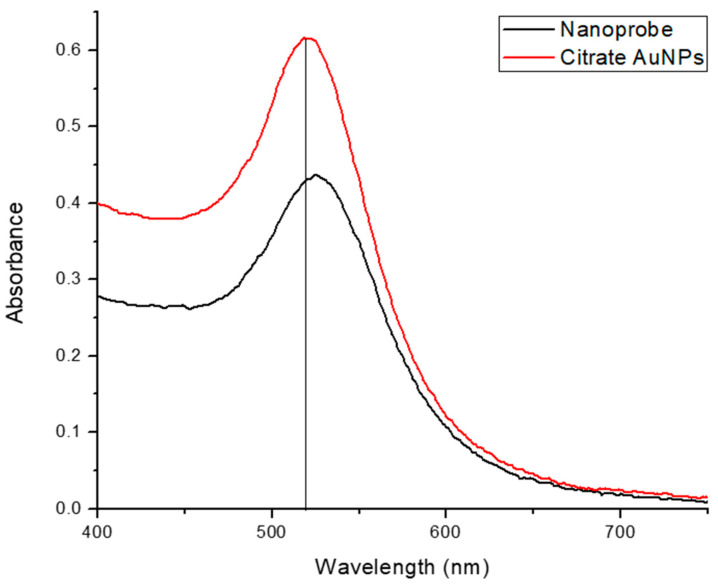
Superimposed spectra of citrate AuNPs and nanoprobe.

**Figure 6 sensors-25-03312-f006:**
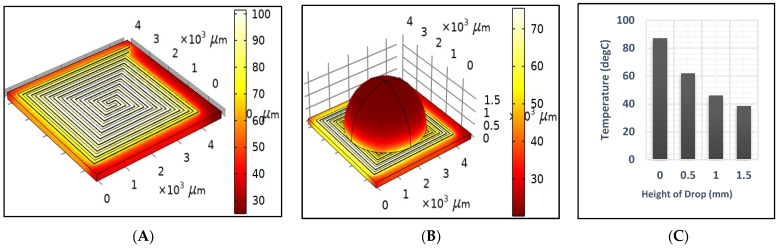
The generated temperature after 1 s using a 2 A current source in the (**A**) dry test (**B**) sample test, and (**C**) temperature versus the height of the drop.

**Figure 7 sensors-25-03312-f007:**
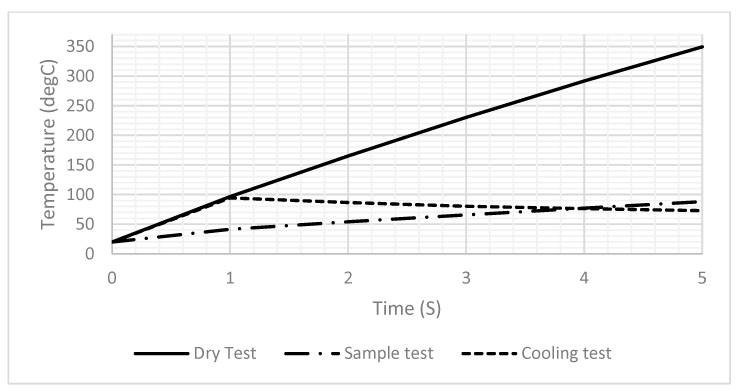
A 1D plot of the generated temperature versus time at the center point of the DFM in the dry test, the sample, and the cooling test.

**Figure 8 sensors-25-03312-f008:**
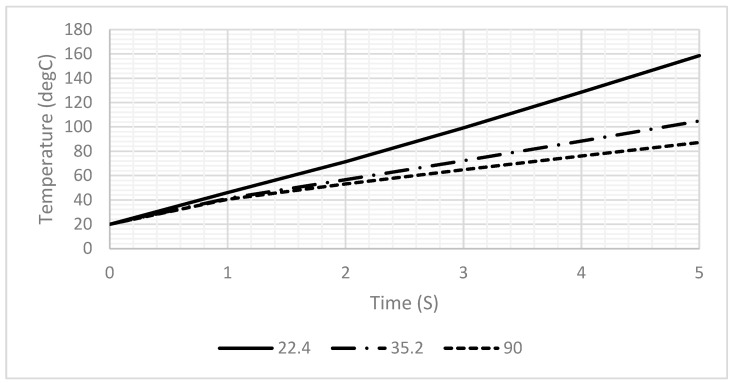
A 1D plot of the generated temperature versus the contact angle of the droplet.

**Figure 9 sensors-25-03312-f009:**
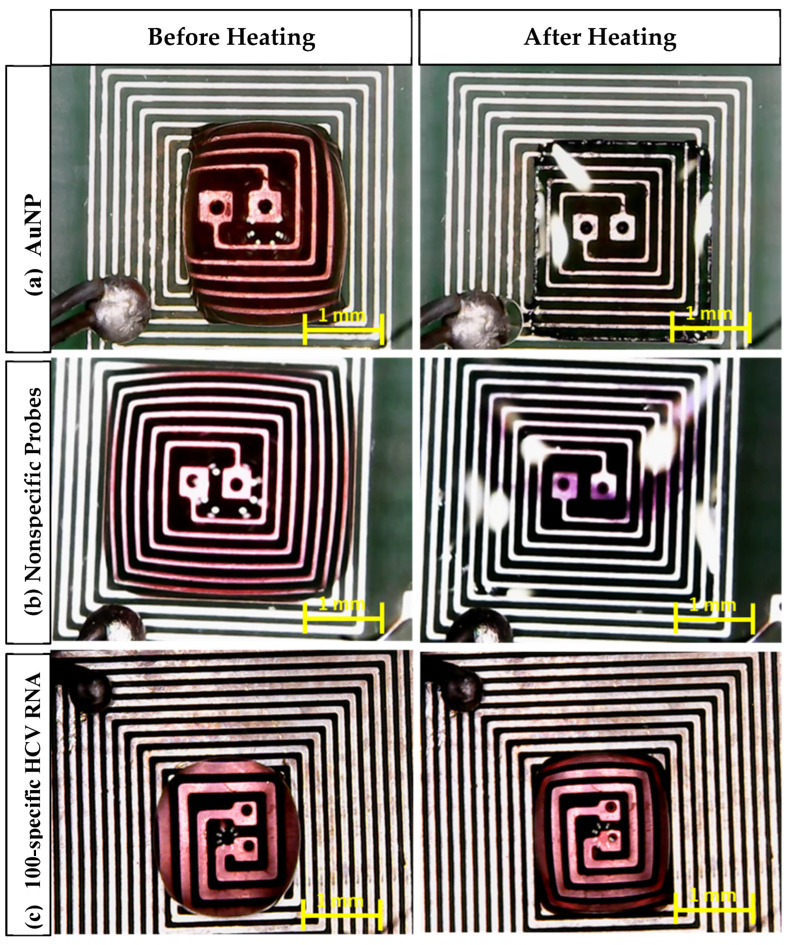
Microscopic images of the aggregation of AuNPs after 105 s of heating: (**a**) AuNPs, (**b**) nonspecific probes, (**c**) 100-specific HCV RNA probes.

**Figure 10 sensors-25-03312-f010:**
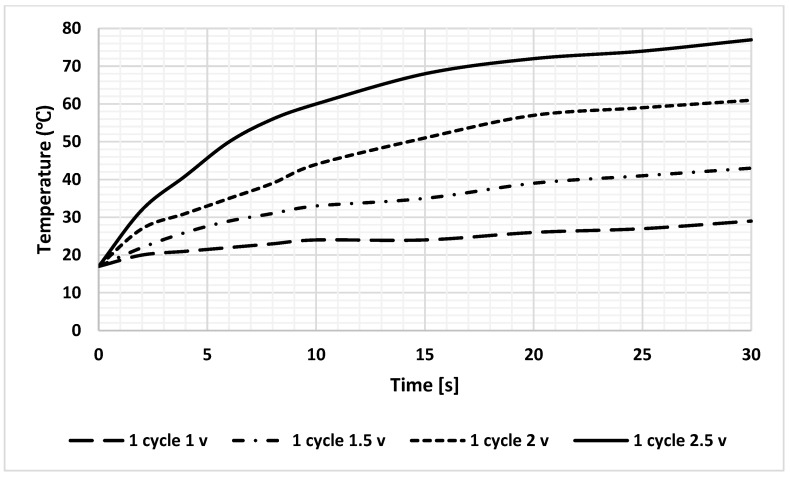
Heating profile of DFM at different DC voltages; the loaded current for all lines is 1.5 A.

**Figure 11 sensors-25-03312-f011:**
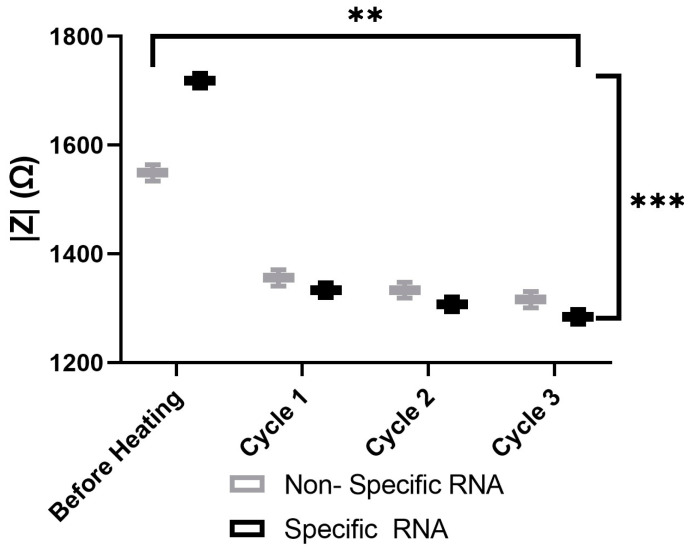
The impedance magnitude of the nonspecific probes and 100-specific HCV RNA probes. Two-way Anova *p*-value = 0.0013 between heating cycles. Where (**) represents a *p*-value of 0.01 or less and (***) represents a *p*-value of 0.001 or less.

**Table 1 sensors-25-03312-t001:** The length and resistance of all proposed designs.

Item	The Total Length (µm)	The Total Resistance (Ω)	
		Mathematical	FEM
DFM	48,400	41.624	38.746

**Table 2 sensors-25-03312-t002:** The grid independence study.

Features	Normal	Fine
Max element size [μm]	480	384
Min element size [μm]	86.4	48
Growth rate	1.5	1.45
Computational time	25 s	3 min 36 s
Readings		
Time [s]	Temp [°C]	Temp [°C]
0	20.00775	20.00937
1	96.48499	96.96841
2	164.9275	165.6715
3	230.1362	231.3351
4	291.6131	293.4089
5	349.301	351.7258
Max Difference %	0.694% at 5 s < 1%	

**Table 3 sensors-25-03312-t003:** Saleae Logic Pro 8 Specifications.

Feature Description	Feature Description
No. of channels	No. of channels: Eight analog inputs (shared with digital channels)
Maximum Sample Rate	Maximum Sample Rate Analog: 50 MSPS @ 3 channels, 12.5 MSPS @ 8 channels
Analog Resolution	Analog Resolution 12 bits, 4.88 mV per LSB
Analog Input Range	Analog Input Range –10 V to 10 V
Capture Buffer	Capture Buffer Length is limited by installed memory. When recording analog at 50 MSPS, captures 10–60 s.
Analog Bandwidth	Analog Bandwidth (–3 dB) 5 MHz

**Table 4 sensors-25-03312-t004:** Comparison with previous work.

	Shape	Size [μm]	Voltage	Heating
[22]	triangle	250	1400 V AC	70 °C in 1 min
This work	spiral	100	2.5 V DC	75 °C in 0.5 min

**Table 5 sensors-25-03312-t005:** Comparison with traditional methods.

Aggregation	Time	Portability	Power	Prescision	Cost
Laser [39,40]	30–40 min	Low	5 watt	High	High
Salt [41]	varies	High	moderate	Moderate	Low
Water [42]	varies	Moderate	varies	Moderate	Moderate
This work Joule heating	2 min	High	3.75 watt	High	Low

## Data Availability

The datasets used and analyzed during the current study are available from the corresponding author upon reasonable request.

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
