# Peer review of "A Novel Microelectrode Based on Joule Heating and Impedance Spectroscopy for Inducing and Monitoring the Aggregation of HCV-Specific Probes"

_sensors, 2025, doi:10.3390/s25113312_

Round 1
Reviewer 1 Report
Comments and Suggestions for Authors
In this paper, a dual-function microelectrode based on Joule heating and impedance spectroscopy is proposed to induce and monitor the aggregation of gold nanoparticles in real time, so as to achieve efficient detection of hepatitis C virus (HCV) RNA. I think authors need to address the following issues:
1. In the abstract, it is suggested to add important quantitative conclusions.
2. In the introduction part, the author did not thoroughly analyze the limitations of existing temperature-induced aggregation technologies (such as laser and water bath), resulting in insufficient demonstration of the advantages of joule heating. In addition, the authors do not clearly explain why existing impedance spectroscopy techniques cannot monitor the aggregation process in real time, and how dual-function microelectrodes in this paper fill this gap. It is suggested that the author supplement the above contents to enhance the innovation of this paper.
3. In the process of numerical modeling, the author has the following deficiencies:
a) How do the boundary conditions correspond to the physical model? Now it's hard to see in Figure 1.
b) The droplet model is hemispherical, but the actual contact angle may affect the temperature distribution, and the rationality of the model hypothesis needs to be explained.
c) Lack of grid independence verification.
4. During the design of the experiment, the author did not explain how to calibrate the accuracy of the external temperature sensor, nor did he provide details of the feedback control algorithm.
5. Figure 6 shows that the temperature drops to 41.5°C in the presence of the droplet, but the experiment uses 2.5V to reach 75°C in 30 seconds, without explaining how the heat loss contradiction is overcome.
6. The impedance spectrum results (Figure. 10) did not provide error analysis, and the reliability of the data could not be evaluated.
7. What is the detection limit of this method? It is suggested that the author compare with existing methods (such as salt-induced aggregation, laser heating, water bath temperature control), and prove that "Joule heating + impedance dual-function integration" is superior to traditional techniques.
In addition, there are many small problems in the full text, it is recommended that the author check the details of the full text. Here are just some of the problems:
1. In the opening paragraph of the methods section, the last sentence is incomplete.
2. On line 236, is Figure 5 represented here?
3. The line spacing of the conclusion should be uniform.
Author Response
Comment 1: In the abstract, it is suggested to add important quantitative conclusions.
Response 1: Thanks for pointing this out. We agree with this comment. Therefore, the abstract of the manuscript has been updated to include quantitative conclusions, as presented on page 2, lines 45–47.
Comment 2: In the introduction part, the author did not thoroughly analyze the limitations of existing temperature-induced aggregation technologies (such as laser and water bath), resulting in an insufficient demonstration of the advantages of joule heating. In addition, the authors do not clearly explain why existing impedance spectroscopy techniques cannot monitor the aggregation process in real time, and how dual-function microelectrodes in this paper fill this gap. It is suggested that the author supplement the above contents to enhance the innovation of this paper.
Response 2: Thanks for pointing this out. We agree with this comment. Therefore, the introduction of the manuscript has been updated to declare these concepts, as presented on pages 2-3, lines 85–92, and lines 120-124.
Comment 3: In the process of numerical modeling, the author has the following deficiencies: a) How do the boundary conditions correspond to the physical model? Now it's hard to see in Figure 1.
Response 3-a: Thanks for pointing this out. We agree with this comment. Therefore, the caption of Figure 2 is updated to include the detailed boundary conditions, as presented on page 6, lines 240–242.
b) The droplet model is hemispherical, but the actual contact angle may affect the temperature distribution, and the rationality of the model hypothesis needs to be explained.
Response 3-b: Thanks for pointing this out. We agree with this comment. Therefore, the caption of Figure 2 is updated to include additional study of three different contact angles, as presented on page 5, lines 219 – 221. In addition, the detailed model hypothesis is described, as presented on page 5, lines 201–208. And the results of the simulation of droplets with different contact angles, as presented on page 12, lines 376–380.
c) Lack of grid independence verification.
Response 3-c: Thanks for pointing this out. We agree with this comment. Accordingly, a meshing section has been included to specify the type of element size and the criteria for its selection, as presented on page 7, lines 262–268.
Comment 4: During the design of the experiment, the author did not explain how to calibrate the accuracy of the external temperature sensor, nor did he provide details of the feedback control algorithm.
Response 4: Thanks for pointing this out. We agree with this comment because it is very important to declare the method of measuring temperature. However, the temperature was measured using a Fluke 87V Industrial Multimeter, which had been calibrated by the manufacturer; further details are provided in the manuscript, as presented on page 9, lines 307–311.
Comment 5: Figure 6 shows that the temperature drops to 41.5°C in the presence of the droplet, but the experiment uses 2.5V to reach 75°C in 30 seconds, without explaining how the heat loss contradiction is overcome.
Response 5: Thanks for pointing this out. We agree with this comment. As illustrated in the figure depicting temperature measurements from the experimental work, This concept is addressed by applying the maximum feasible voltage within an appropriate duration, determined through extensive trials involving various voltage levels and heating durations, as presented on page 13, lines 397–403.
Comment 6: The impedance spectrum results (Figure. 10) did not provide error analysis, and the reliability of the data could not be evaluated.
Response 6: Thanks for pointing this out. We agree with this comment. The figure is updated to declare the error analysis. In addition, the reliability of data is performed, as presented on page 14, lines 415–425.
Comment 7: What is the detection limit of this method? It is suggested that the author compare with existing methods (such as salt-induced aggregation, laser heating, water bath temperature control), and prove that "Joule heating + impedance dual-function integration" is superior to traditional techniques.
Response 7: Thanks for pointing this out. We agree with this comment. A declaration of the optimized as minimum nucleotide length and concentration of target RNA, as presented on page 10, lines 346– 349. In addition, Table 5 and the summary are added for a comparison between the current methods and traditional methods in terms of time of test, portability, power, precision, and cost, as presented on page 10, lines 433– 448.
In addition, there are many small problems in the full text, it is recommended that the author check the details of the full text. Here are just some of the problems:
1. In the opening paragraph of the methods section, the last sentence is incomplete.
2. On line 236, is Figure 5 represented here?
3. The line spacing of the conclusion should be uniform.
Response 8: Thanks for pointing this out. The entire article has been revised, and the indicated points have been addressed.
Reviewer 2 Report
Comments and Suggestions for Authors
The article presents the development of an innovative microelectrode integrated into a chip, capable of inducing and monitoring the aggregation of gold nanoparticles (AuNPs) attached to RNA molecules specific to the Hepatitis C Virus (HCV). This device has a dual function: it generates heat through the Joule effect and measures the electrical properties of the sample using impedance spectroscopy. The rapid and accurate detection of mutant viruses, such as HCV, is essential in the current global context. Gold nanoparticles have unique optical properties, and when they aggregate, they cause a visible color change in the solution (e.g., from red to violet), which can be exploited for diagnostic purposes. However, some minor revisions are recommended before publication:
The introduction provides a solid context regarding the need for new technologies in virus detection, particularly for HCV. It explains the fundamentals related to gold nanoparticles and aggregation methods. Relevant and up-to-date references are included (up to 2025), covering topics such as AuNPs, aggregation, optical detection, and impedance. However, the introduction could be strengthened by including more direct comparisons with existing similar technologies ( portable biosensors, integrated microelectrodes).
The research design is well thought out and innovative. The combination of: Joule heating,impedance spectroscopy,and PCB integration provides a solid and practical experimental framework for testing hypotheses related to nanoparticle aggregation in the detection of HCV RNA.
The methods are described in detail. The results are clearly presented through: temperature-over-time graphs, FEM simulations (heat distribution),microscopic images of aggregation, and impedance value plots. The presentation is clear and supported by explanations. However, it would be helpful if some figures were numbered and discussed more explicitly in the text:
Figure 6 and Figure 7
Both figures are mentioned, but there are no explicit references to the subfigures in the text (e.g., "Fig. 6 (a), (b), (c)" appears in the caption, but the text does not clearly explain what each subfigure represents).
It would be helpful if the text detailed: "In Fig. 6(a)..." / "Fig. 6(b) shows..." etc.
The conclusions are logically supported by the obtained results, both simulated and experimental.
Overall, this is a strong and original contribution that is suitable for publication after minor revisions.
Author Response
Comment 1: Both figures are mentioned, but there are no explicit references to the subfigures in the text (e.g., "Fig. 6 (a), (b), (c)" appears in the caption, but the text does not clearly explain what each subfigure represents). It would be helpful if the text detailed: "In Fig. 6(a)..." / "Fig. 6(b) shows..." etc.
Response 1: We appreciate your valuable comment and fully agree with it. The manuscript has been updated accordingly, as detailed on page 10, lines 353–355.
Round 2
Reviewer 1 Report
Comments and Suggestions for Authors
The author's revisions have solved all my problems. This work can be published!